# The Role of Pyk2 Kinase in Glioblastoma Progression and Therapeutic Targeting

**DOI:** 10.3390/cancers17162611

**Published:** 2025-08-09

**Authors:** Lilia Kucheryavykh, Yuriy Kucheryavykh

**Affiliations:** Department of Biochemistry, Universidad Central del Caribe, Bayamón, PR 00956, USA; lilia.kucheryavykh@uccaribe.edu

**Keywords:** Pyk2, GBM, targeted therapies, combination therapies

## Abstract

Glioblastoma is an aggressive brain tumor with limited treatment options and a poor outcome. Proline-rich tyrosine kinase 2 (Pyk2) is involved in glioblastoma proliferation, invasion, and recurrence. Cytokines and chemokines, released from tumor-infiltrating myeloid cells, are involved in activation of Pyk2 signaling in glioma cells. Recurrent tumors, as well as temozolomide-treated tumors, show higher Pyk2 phosphorylation in animal models. Targeting Pyk2 has shown promise in reducing recurrence and improving survival in preclinical models. This review highlights Pyk2 as a potential treatment target for glioblastoma.

## 1. Introduction

Glioblastoma (GBM) is the most common and aggressive primary brain tumor in adults, with a median survival of approximately 12 to 15 months and an overall poor prognosis [1,2,3]. Despite advancements in healthcare, the five-year survival rate remains below two percent, primarily due to the highly invasive nature of GBM cells, which leads to rapid recurrence after surgical resection [4,5]. A major challenge in treating GBM is the tumor’s strong resistance to standard therapies, including both radiation and chemotherapy. This resistance significantly limits the effectiveness of temozolomide (TMZ), the current standard-of-care chemotherapeutic agent used in GBM treatment [6]. Resistance to alkylating agents like TMZ arises through multiple mechanisms. One major route involves the repair of TMZ-induced DNA damage: methylated DNA lesions can be corrected by base excision repair (BER) and DNA mismatch repair (MMR) pathways. Additionally, the DNA repair enzyme O6-methylguanine-DNA methyltransferase (MGMT) can directly reverse methylation at the O6 position of guanine, a key cytotoxic lesion induced by TMZ. Beyond DNA repair, resistance is also driven by the dysregulation of critical signaling pathways—such as Akt, Wnt/β-catenin, and JAK/STAT—that regulate cell cycle progression, apoptosis, and cellular metabolism [7,8,9,10,11]. Given these resistance mechanisms, targeting alternative molecular pathways may offer a promising strategy to improve therapeutic outcomes in glioblastoma.

Proline-rich tyrosine kinase 2 (Pyk2) is a non-receptor tyrosine kinase that serves as an important mediator of intracellular signaling, linking extracellular stimuli to intracellular pathways that regulate cell proliferation, migration, and survival (Figure 1) [12,13,14,15,16]. Pyk2 is also activated in response to elevated intracellular calcium levels and is involved in several downstream signaling cascades, including the Ras, MAP kinases, and NF-kB pathways [17,18,19]. Elevated Pyk2 activity has been associated with increased tumor cell migration and invasion, identifying it as a critical contributor to GBM progression [16,20,21,22]. Studies have shown that Pyk2 overexpression enhances tumor cell dispersal, whereas its inhibition reduces invasiveness and improves survival in glioma-bearing mouse models [23,24,25].

In addition to its intrinsic roles within tumor cells, Pyk2 activity is strongly influenced by the tumor microenvironment. Tumor-associated myeloid cells and microglia (TAMs) (key components of the GBM microenvironment) contribute to GBM progression by secreting cytokines and chemokines, such as PDGFB, EGF, SDF-1α, IL-6, and IL-8 (Figure 2). These soluble factors activate Pyk2 signaling in GBM cells, further promoting tumor cell migration and invasion [14,15,20,22,23,26]. Pyk2 functions as a signaling hub [27,28,29,30,31,32], integrating inputs from growth factor receptors, G protein-coupled receptors, interleukin 6 receptor, and integrins, thereby enabling tumor cells, as well as normal brain astrocytes and neurons, to respond dynamically to their surroundings. This makes Pyk2 an important mediator of microglia-stimulated glioma cell motility and GBM tumor invasiveness. Importantly, by facilitating tumor adaptation to environmental and therapeutic stressors, Pyk2 signaling may also contribute to resistance against conventional therapies, including temozolomide and radiation, further underscoring its relevance as a therapeutic target in GBM.

Given the multifaceted role of Pyk2 in mediating glioblastoma progression—through both intrinsic tumor cell signaling and extrinsic interactions with the tumor microenvironment, this review aims to provide a comprehensive overview of Pyk2’s biological functions, regulatory mechanisms, and involvement in GBM pathophysiology. We also highlight emerging evidence supporting Pyk2 as a potential therapeutic target, discussing its role in treatment resistance and tumor invasiveness. By synthesizing current findings, this manuscript seeks to inform future research directions and explore the therapeutic potential of targeting Pyk2 in the context of overcoming GBM’s resistance to standard therapies.

## 2. The Role of Pyk2 in Microglia-Induced Glioma Invasion

Previous findings [20] highlight the significance of Pyk2 in glioma cell migration and invasion. In vitro experiments using several established glioma cell lines (A172, U87, HS683 and GL261) demonstrated that cells with higher levels of Pyk2 expression and phosphorylation exhibited greater invasive capacity, whereas those with lower basal Pyk2 activity showed limited migration and invasiveness. Pharmacological inhibition of Pyk2 reduced basal migration in U87 and HS683 human glioma cells but had little to no effect on A172 human, GL261 mouse, or C6 rat glioma cells suggesting cell line-specific dependencies on Pyk2 signaling.

Notably, treatment of glioma cells with medium, conditioned from microglia (MCM), led to increased phosphorylation of Pyk2 at Tyr 579/580 and significantly enhanced invasiveness, even in cell lines typically characterized by low invasive potential, such as HS683. Inhibition of Pyk2, either pharmacologically using Pyk2/FAK inhibitor PF-562,271 (13 nM) or via siRNA-mediated knockdown, effectively blocked microglia-induced glioma cell migration. These results strongly support the role of Pyk2 in microglia-stimulated glioma invasiveness. In vivo studies using the CD11b-HSVTK/ganciclovir model (a syngeneic system involving intracranial implantation of GL261 glioma cells into mice, engineered to express herpes simplex virus thymidine kinase (HSV-TK) under the CD11b promoter) provided further validation. Selective depletion of microglia in this model via ganciclovir treatment significantly reduced Pyk2 phosphorylation in tumor tissue and was associated with a less invasive glioma phenotype, compared to untreated controls [20]. These findings suggest that the activation of the Pyk2 signaling in glioma cells, driven by interactions between glioma and TAM/microglia, promotes tumor cell dispersal and invasion into surrounding brain parenchyma.

Microglia and TAMs play an important role in glioma progression by shaping the tumor microenvironment [33,34,35,36,37,38,39]. Rather than mounting an effective immune response, these cells secrete a variety of cytokines, chemokines, and proteolytic enzymes, including matrix metalloproteinases (MMPs) and cathepsins, that degrade the extracellular matrix (ECM), facilitating tumor invasion and supporting glioma cell survival and proliferation [40,41,42,43,44,45,46,47,48].

Beyond Pyk2, other signaling molecules are implicated in glioma pathogenesis. For instance, insulin-like growth factor binding protein 3 (IGFBP-3), Cullin1 (Cul1), Golgi phosphoprotein 3 (GOLPH3), and aquaporin 1 (AQP1) have all been associated with glioma progression and invasiveness [49,50,51,52]. Interestingly, although the structurally related focal adhesion kinase (FAK) shares signaling overlap with Pyk2, its role in glioma migration remains uncertain. Some studies, including those by Lipinski [16] and others using FAK inhibitors [20], report little to no effect of FAK inhibition on glioma migration across multiple cell lines, regardless of microglial presence. However, contrasting reports suggest that FAK may contribute to glioma invasiveness under certain conditions [53,54], highlighting the need for further investigation into the distinct and overlapping roles of these kinases in GBM biology.

## 3. Regulation of Glioma Cell Invasion and Proliferation by Microglia-Derived Cytokines and Chemokines Through Pyk2 Signaling

To better understand how microglia-derived cytokines and chemokines activate Pyk2 signaling in glioma cells, studies were performed using microglia treated with glioma-conditioned medium (GCM) derived from primary glioma cell lines [22]. Exposure to GCM significantly enhanced the secretion of key pro-tumorigenic factors by microglia, including PDGFB, EGF, SDF-1α, IL-6, and IL-8, in vitro. When glioma cells were subsequently treated with conditioned medium from glioma-microglia co-cultures (in a 3:1 ratio, reflecting cellular proportions found in GBM tumors), Pyk2 phosphorylation in glioma cells was robustly induced.

Among these factors, IL-6 and EGF consistently promoted Pyk2 activation across all glioma cell lines tested, regardless of their baseline levels of total Pyk2 expression. In contrast, PDGFB, SDF-1α, and IL-8 exhibited patient-specific effects, suggesting variability in glioma subtype responsiveness. Importantly, pharmacological inhibition or genetic silencing of the receptors for these cytokines abolished microglia-induced Pyk2 phosphorylation, confirming that microglia-derived factors are essential for Pyk2 signaling in glioma cells.

### 3.1. Pyk2 and Invadopodia Formation

Pyk2 regulates the formation of invadopodia [22], specialized actin-rich structures used by tumor cells to degrade the extracellular matrix and invade surrounding tissue. Microglia-derived PDGFB and SDF-1α significantly enhanced invadopodia activity in primary GBM cells, particularly in glioma subtypes expressing moderate levels of total Pyk2. This effect was strictly Pyk2-dependent. These findings are consistent with prior studies [55] showing that CXCL12 (SDF-1α) promotes invadopodia formation through cortactin-mediated actin polymerization, a pathway in which Pyk2 is known to play a critical role.

### 3.2. Pyk2-Dependent Cell Migration

Further evidence of Pyk2′s involvement in glioma motility came from cell migration assays. PDGFB, EGF, and SDF-1α all promoted glioma cell migration in a Pyk2-dependent manner, as their effects were abolished upon Pyk2 knockdown [22]. Additionally, primary glioma cell lines with low expression of total Pyk2 exhibit autocrine SDF-1α expression, suggesting a self-sustaining mechanism that supports continued tumor cell invasion. In contrast, glioma cell lines with high Pyk2 expression may rely on additional or alternative signaling pathways beyond Pyk2 and FAK for migration in response to SDF-1α and EGF, although Pyk2 still appears to act as a central regulator of cell dispersal [20,53,54,56].

### 3.3. Pyk2 and Glioma Proliferation

Beyond its role in migration, Pyk2 also contributes to glioma cell proliferation. Glioma cells secrete various factors, including MCP-1, GDNF, SDF-1, that recruit TAMs to the tumor microenvironment. In turn, TAMs promote glioma growth and proliferation by releasing a range of cytokines and chemokines, establishing a dynamic crosstalk between the two cell types [35,36]. Exposure of primary human glioma cells to microglia-conditioned medium in vitro has been shown to stimulate glioma cell mitosis, particularly enhancing progression into the G2/M phase of the cell cycle [22]. This effect is mediated largely through Pyk2 signaling in response to factors such as EGF, IL-6, SDF-1α, and PDGFB. Notably, IL-8 was also found to regulate mitosis, but through a Pyk2-independent mechanism, suggesting that multiple, parallel signaling pathways may coordinate glioma growth [22].

Collectively, these studies underscore the role of Pyk2 in microglia-induced glioma progression. Microglia-derived cytokines and chemokines, particularly PDGFB, EGF, SDF-1α, IL-6, and IL-8, activate Pyk2, but only PDGFB, EGF, SDF-1α and IL-6 promote Pyk2-related glioma cell invasion and proliferation. The differential activation of Pyk2 across glioma subtypes reflects the complexity of tumor–TAM interactions and suggests subtype-specific therapeutic vulnerabilities. Future research should aim to define the broader signaling networks linking microglia to Pyk2 activation and evaluate targeted strategies to disrupt this pro-tumorigenic axis.

## 4. Pyk2 and GBM Recurrence

A major challenge in the treatment of primary GBM is its high recurrence rate, which significantly contributes to its poor prognosis. Despite aggressive multimodal treatment, including surgical resection followed by chemoradiotherapy, tumor relapses are almost inevitable. The mechanisms driving this recurrence remain incompletely understood.

Recent evidence, using the GL261/C57Bl/6 mouse glioma implantation model, has demonstrated a significant increase in Pyk2 signaling activity in regrown tumors after surgical resection, despite no difference in the basal expression levels of total Pyk2 when compared to primary implanted untreated tumors [24]. This suggests that Pyk2 activation is dynamically regulated during recurrence and may not depend solely on expression levels, but rather on upstream environmental cues and phosphorylation events. Given the established role of Pyk2 in glioma proliferation and invasion, its post-surgical activation may directly contribute to tumor regrowth and affect patient survival.

Pharmacological inhibition of Pyk2/FAK signaling using PF-562271 (25 mg/kg, twice daily) significantly suppressed Cyclin D1 expression in recurrent tumors, indicating disruption of cell cycle progression [24]. This effect aligns with previous reports showing that Pyk2 inhibition can destabilize β-catenin and suppress downstream oncogenic targets such as c-Myc and Cyclin D1, thereby interfering with the Wnt/β-catenin signaling pathway [57,58]. Additionally, a reduction in the Ki67 proliferation index following PF-562271 treatment supports the role of Pyk2 in regulating tumor cell mitosis and reinforces its involvement in GBM regrowth and glioma cell proliferation [24]. These results support prior studies demonstrating the efficacy of Pyk2 inhibition in reducing tumor growth and invasion across various cancer models, including breast cancer and GBM [59,60,61].

These findings are consistent with broader cancer research. For instance, in osteosarcoma xenograft models, PF-562271 treatment reduced tumor weight and volume [57], highlighting the importance of Pyk2 in promoting tumor invasiveness and proliferation. In GBM specifically, studies using both primary human GBM cells and the GL261/C57Bl/6 mouse GBM model have further demonstrated that Pyk2 inhibition not only reduces cell proliferation but also enhances the efficacy of temozolomide, the standard chemotherapeutic agent [25]. The combination therapy produced greater apoptosis, deeper cell cycle arrest, and reduced invasion compared to temozolomide monotherapy.

In the context of recurrent GBM, PF-562271 treatment led to a 43% reduction in tumor volume in post-resected regrown tumors and increased the median survival rate by 33% compared to the untreated group [24]. These findings suggest that targeting Pyk2 signaling may be a promising therapeutic target for managing GBM recurrence, particularly for residual, invasive tumor cells that survive surgical removal and initiate recurrence. Effective targeting of these cells requires a deeper understanding of the distinct molecular and cellular characteristics of the invasive tumor margins versus the core [62].

Notably, recurrent tumors often emerge in a tumor microenvironment altered by treatment. Therapeutic interventions such as chemotherapy and radiation contribute to an acidic microenvironment, which has been shown to enhance tumor invasion via acid-sensing ion channels (ASICs). These channels increase intracellular calcium and promote epithelial–mesenchymal transition (EMT) [63,64]. Pyk2, being a pH-sensitive kinase, is activated under acidic conditions in several cancers, implying that post-treatment acidosis may drive Pyk2-mediated tumor progression in GBM [65].

Changes in the immune microenvironment also contribute to recurrence. Studies have reported distinct differences in tumor-associated myeloid cells (TAMs) between primary and recurrent human GBM specimens [66,67]. While both primary and recurrent tumors exhibited similar levels of myeloid and T cell infiltration, there were notable shifts in TAM activation states and spatial organization [68]. Specimens from newly diagnosed GBMs were predominantly infiltrated by TAMs with microglial gene signatures, whereas recurrent tumors exhibited a transition toward monocyte-derived TAMs, which have been linked to poor survival outcomes in GBM patients [68,69].

In studies using a Gl261/C57Bl/6 mouse GBM model, no significant differences in the ratio of peripheral macrophages (CD45^high^) to microglia (CD45^low^) between primary and recurrent tumors were found, suggesting that TAM infiltration remained stable [66]. However, significant changes in TAM polarization and cytokine expression were reported. Primary implanted tumors were characterized by a CD206^+^/CD86^−^ TAM population, while recurrent tumors exhibited a CD206^+^/CD86^+^ phenotype. This shift was accompanied by upregulated expressions of IL4, IL5, IL10, IL12, IL17, vascular endothelial growth factor (VEGF), and monocyte chemoattractant protein 1 (MCP1/CCL2) in TAM cells from recurrent tumors, compared with primary implanted tumors. TAM-derived cytokines and growth factors such as EGF, PDGF, SDF1, IL8, and IL6 have been implicated in activating Pyk2 signaling, and these findings suggest that the shift in TAM polarization in recurrent GBM may contribute to the observed upregulation of Pyk2 phosphorylation in GBM cells [25].

Consistent with this, previous work using CD11b-HSVTK transgenic mice demonstrated that microglia and macrophage depletion significantly reduced pPyk2 (579/580) expression in glioma cells [20]. Moreover, PF-562271 not only inhibited Pyk2 signaling in tumor cells but also reduced TAM infiltration in recurrent tumors [24], suggesting a dual mechanism of action: direct inhibition of tumor cell signaling and disruption of TAM-mediated Pyk2 activation. Given that Pyk2 is also expressed in myeloid cells [67], it is possible that PF-562271 directly limits TAM migration and accumulation within the tumor resection site. This dual effect further underscores its therapeutic potential in the recurrent setting. However, due to genetic and phenotypic heterogeneity of GBM, further validation across multiple animal models and patient-derived systems is essential.

Finally, as evidence accumulates that PF-562271 enhances temozolomide efficacy [25], future work should prioritize combinatorial strategies that target both tumor-intrinsic and microenvironmental drivers of recurrence. Recurrent GBM remains a clinical challenge, and therapies that interrupt both the tumor and its supporting niche offer the most promise. In conclusion, targeting Pyk2 signaling in the recurrent GBM setting may provide significant therapeutic benefits. By targeting tumor proliferation, limiting immune cell-mediated tumor support, and sensitizing tumors to standard treatments, Pyk2 inhibitors represent a compelling addition to the arsenal against GBM recurrence.

## 5. Pyk2 and Resistance to TMZ

Resistance to TMZ is a major clinical obstacle. This resistance varies across different tumors and can be categorized as either innate or acquired, often involving alterations in DNA repair genes [70,71,72]. Enhanced Pyk2 signaling was observed following TMZ treatment [25], indicating that this pathway may contribute to TMZ resistance. These findings suggest that inhibiting Pyk2 signaling could potentially improve treatment effectiveness.

### 5.1. Enhanced Cytotoxicity Through Pyk2 Inhibition

Pharmacological inhibition of Pyk2/FAK signaling using PF-562271 has been shown to enhance TMZ-induced cytotoxicity in both primary human GBM cells and the GL261 murine glioma model [25]. Cell cycle analysis of the primary human GBM cells with low and moderate expression of total Pyk2, and GL261 mouse cells, revealed an increase in the G2/M and sub-G1 populations following 72 h TMZ treatment, along with upregulation of Bcl2 expression, indicating cell cycle arrest. GBM resistance to therapy is frequently associated with the overactivation of G2 checkpoint kinases [73], which facilitate DNA repair and prevent mitotic catastrophe [74,75]. Notably, the combinatorial treatment with TMZ and PF-562271 further increased the sub-G1 population while affecting the G2/M phase differently across these cell lines [25]. In the primary human GBM cells with low expression of total Pyk2 and in GL261 cells, the combination led to a pronounced G2/M arrest, while in cells with moderate Pyk2 expression, this effect was absent. Additionally, Bcl-2 expression was markedly reduced with the combinatorial treatment compared to TMZ alone, indicating that Pyk2 inhibition may reverse TMZ-induced anti-apoptotic signaling.

In cells with low total Pyk2, the combination also led to G1 phase accumulation, implying a potential role for Pyk2 in regulating the S/G2 transition. Although the precise mechanism remains unclear, our findings support a model in which Pyk2 contributes to progression through the S/G2 phase of the cell cycle. TMZ induces G2/M arrest as part of a treatment resistance mechanism. We propose that PF-562271 enhances this arrest by preventing re-entry into the cell cycle, thereby amplifying the cytotoxic effects of TMZ. This supports previous findings [22], which suggest that impairing Pyk2 activity may increase both the cytostatic and cytotoxic effects of TMZ by disrupting cell cycle checkpoints.

### 5.2. Influence of NF1 Status on Treatment Response

Importantly, this effect was not observed in cells with moderate expression of total Pyk2, which lacks neurofibromatosis 1 (NF1). NF1 loss is known to sustain activation of RAS-GTP, leading to persistent stimulation of downstream pathways including RAS/RAF/MAPK, Akt, and FAK, as well as upregulation of HSF1, a factor associated with cell proliferation, survival, and resistance to therapy [76,77,78,79,80]. Prior research suggests that NF1-deficient cancers can be sensitized to therapy through FAK and MAP-ERK inhibition [76,81,82]. Despite these oncogenic drivers, recent data indicates that PF-562271 retains efficacy in both NF1-positive and NF1-deficient GBM models [25], likely due to its broader impact on Pyk2- and FAK-mediated signaling.

### 5.3. Apoptosis and Alternative Death Pathways

While the number of apoptotic GL261 cells in vitro did not significantly differ between the combinatorial treatment and TMZ monotherapy, the overall reduction in total cell count suggests the involvement of alternative cell death mechanisms beyond apoptosis. Previous studies have implicated autophagy as a key pathway in TMZ-induced GBM cell death [83,84]. Moreover, in GL261 cells, the combination treatment was associated with Cyclin D1 upregulation [25], which may play a paradoxical pro-death role under stress conditions. Cyclin D1 overexpression has been linked to enhanced caspase-dependent apoptosis and ER stress-induced cell death in various cancers, though further studies are needed to determine its precise role in GBM treatment susceptibility [85,86].

### 5.4. Effects on Invasion and Invadopodia Activity

Invasion and ECM degradation, facilitated by invadopodia, are critical for GBM progression [87]. While TMZ/PF-562271 combinatorial treatment did not significantly impact cell migration relative to TMZ alone, it did lead to a marked reduction in invadopodia activity and invasion across cells with low and moderate expression of total Pyk2, and GL261 [25]. These findings align with previous reports that GBM cells acquire a more invasive phenotype following TMZ treatment [88,89]. Given that Pyk2 is an important regulator of invasion and extracellular matrix degradation [16,20], its inhibition may counteract the pro-invasive remodeling induced by TMZ. Notably, the strongest anti-invasive response to PF-562271 was observed in cells with moderate Pyk2 expression, potentially reflecting NF1 loss–mediated hyperactivation of Pyk2 [79].

### 5.5. In Vivo Efficacy of the Combinatorial Approach

In vitro findings were further supported by in vivo studies [25]. While TMZ alone (50 mg/kg daily, orally) significantly reduced tumor core volume, it had little effect on the invasive tumor margins. In contrast, PF-562271 alone did not reduce the volume but did attenuate invasion margins. The combinatorial treatment led to reduction in both tumor size and invasive spread, accompanied by decreased Ki67 proliferation, increased apoptosis, and improved survival outcomes. These findings suggest that combining Pyk2/FAK inhibition with TMZ may address two key challenges in GBM treatment: tumor regrowth and invasion.

### 5.6. Clinical Implications and Future Directions

Although Pyk2 inhibitors have demonstrated promising preclinical efficacy in various preclinical models and non-brain tumors [57,90,91], clinical trials using Pyk2/FAK inhibitors as monotherapies have shown only modest benefits in progression-free survival [92,93]. The current evidence indicates that monotherapy may be insufficient, but combinatorial regimens with chemotherapeutics like TMZ could represent a more effective strategy for overcoming resistance and improving clinical outcomes in GBM. In summary, Pyk2/FAK signaling contributes to both TMZ resistance and GBM invasiveness. Inhibiting this pathway with PF-562271 not only enhances TMZ cytotoxicity but also mitigates therapy-induced invasion, offering a multifaceted approach to combat GBM progression. Further studies, including well-designed clinical trials—are warranted to validate the efficacy and optimize the timing and dosing of Pyk2-targeted combination therapies.

## 6. Ethnicity and Sex-Specific Variability in Pyk2 Signaling

Recent findings [94] build upon earlier studies [95,96,97] that underscore the influence of ethnicity and biological sex on GBM pathogenesis, with a particular focus on the role of Pyk2 gene expression (PTK2B) in Puerto Rican patients. By analyzing ethnicity-based variations in gene expression, the authors identified significant correlations involving Pyk2 (PTK2B) alongside other key signaling molecules, including nerve growth factor receptor (NGFR), platelet-derived growth factor receptor β (PDGFRβ), and epithelial growth factor receptor (EGFR), in both male and female Puerto Rican patients. Notably, C-X-C motif chemokine receptor 1 (CXCR1) exhibited a strong correlation with Pyk2 exclusively in Puerto Rican men, a pattern absent in Puerto Rican women and non-Hispanic patients. These findings suggest a sex-specific and ethnicity-dependent regulatory mechanism that may shape Pyk2-driven tumor biology.

### 6.1. Sex Hormones and Differential Pyk2 Signaling

The observed sex-based differences in Pyk2 correlations raise intriguing questions about the underlying molecular mechanisms. The absence of a strong correlation between Pyk2 and CXCR1 in Puerto Rican women suggests a potential hormonal influence. Prior studies indicate that estrogen may play a protective role in GBM [98,99], while androgens and androgen receptors have been shown to influence neural gene expression and may promote oncogenesis [100,101]. These hormonal differences may influence the expression or activity of Pyk2 and its downstream effectors, potentially explaining the sex-specific correlation patterns, such as the exclusive association between Pyk2 and CXCR1 observed in Puerto Rican men. Given that Pyk2 regulates multiple oncogenic pathways, including cell adhesion, migration, and proliferation, understanding how sex hormones affect its expression and function could reveal sex-specific mechanisms of tumor progression and therapy response.

### 6.2. The Genetic Landscape of the Puerto Rican Population

The unique genetic composition of the Puerto Rican population, shaped by a blend of Native American, European, and African ancestries [102], contributes to a distinct genomic landscape. Previous studies have demonstrated that Puerto Rican patients exhibit distinct molecular signatures compared to other Hispanic subgroups and non-Hispanic populations [103,104,105]. For example, EGFR mutations, frequently associated with GBM pathogenesis, have been reported at higher rates in Puerto Rican patients than in Caucasians [106,107]. Since Pyk2 acts downstream of EGFR, these population-specific genetic variations may influence how Pyk2 contributes to tumor biology and therapy response.

While there are no singularly defined “sex-” or “ethnicity-specific” genes shaping Pyk2 function per se, the findings suggest that gene-environment interactions, hormonal regulation, and population-specific genetic variation may collectively create distinct regulatory contexts for Pyk2 activity in GBM.

### 6.3. Toward Precision Medicine: Future Research Directions

These findings underscore the need for a multi-omics approach, incorporating genomics, transcriptomics, and proteomics, to investigate how ethnicity and sex influence Pyk2 signaling and GBM progression. Given prior evidence that targeting Pyk2/FAK signaling enhances the efficacy of temozolomide in GBM models, investigating ethnicity-specific responses to Pyk2 inhibitors could inform the development of personalized therapeutic strategies.

Incorporating ethnic and genetic diversity into future research may refine the understanding of Pyk2 as a therapeutic target and support the development of precision medicine approaches to improve outcomes for GBM patients worldwide.

## 7. Conclusions

Pyk2 kinase is an important regulator of GBM progression, playing a multifaceted role in tumor invasion, proliferation, recurrence, and therapeutic resistance. Its activation, often triggered by tumor-associated microglia and macrophages derived cytokines such as EGF, IL-6, and SDF-1α, promotes a tumor-supportive microenvironment by enhancing glioma cell motility, invadopodia activity, and cell cycle progression. Preclinical studies, both in vitro and in vivo, consistently demonstrate that targeting Pyk2—particularly through the dual Pyk2/FAK inhibitors—can significantly suppress glioma invasion and proliferation, reduce tumor regrowth following surgical resection, and improve the cyto-toxic efficacy of TMZ, especially in TMZ-resistant or NF1-deficient tumors.

Additionally, recent evidence indicates that Pyk2 signaling regulation may vary based on biological sex and ethnicity, highlighting the potential benefit of tailoring Pyk2-targeted therapies through precision medicine approaches. These variations emphasize the importance of incorporating genetic and demographic diversity into future GBM research.

Overall, the data supports the therapeutic potential of Pyk2 inhibition in GBM and provides a strong rationale for further clinical investigation. Pyk2 inhibitors, particularly when used in combination with existing standard-of-care treatments, may offer a promising strategy to improve outcomes in patients with this aggressive and currently incurable brain tumor.

## Figures and Tables

**Figure 1 cancers-17-02611-f001:**
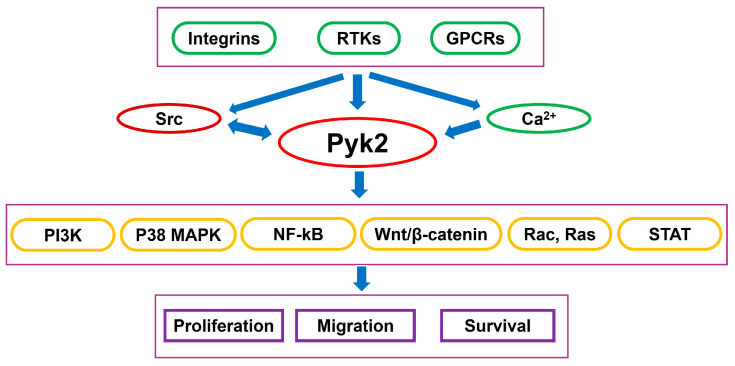
Pyk2 coordinates the transmission of extracellular signals—such as cytokines and growth factors—into intracellular responses that drive glioma cell migration, invasion, and survival.

**Figure 2 cancers-17-02611-f002:**
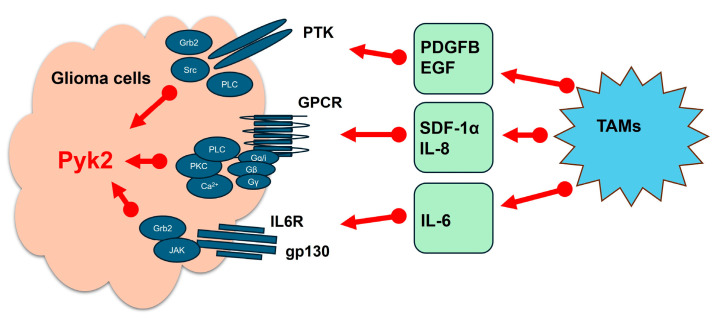
Factors, released from tumor-associated myeloid cells and microglia, that activate Pyk2.

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
