# Peer review of "The Role of Pyk2 Kinase in Glioblastoma Progression and Therapeutic Targeting"

_cancers, 2025, doi:10.3390/cancers17162611_

Round 1

Reviewer 1 Report

Comments and Suggestions for Authors

In their manuscript "The Role of Pyk2 Kinase in Glioblastoma Progression and Therapeutic Targeting" Lilia and Yuriy Kucheryavykh present a review about Pyk2 Kinase in glioblastoma. The review summarizes the actual knowledge about Pyk2 in GBM. Although it is not a systematic review, this methodological weakness is acceptable, since very little is known about Pyk2 in GBM. In this respect, much of the role of Pyk2 in GBM is speculative and not always conclusive. The focus on Puerto Rico, with speculations about hereditary and gender factors involved in differential expression in GBM, stems from the fact that the few articles available on this topic were co-published by the authors of this review. Overall, the work is worth publishing. The only thing that urgently needs to be addressed is an update of the literature, which only includes the authors' own works from the last 3 years, all other articles are older.

Author Response

Comments 1: The only thing that urgently needs to be addressed is an update of the literature, which only includes the authors' own works from the last 3 years, all other articles are older.

Response 1: Thank you for your kind response. Unfortunately, there is limited published data on Pyk2 in glioma; however, we have included several recent articles addressing Pyk2 signaling (ref. 27-32, 61) and update figure 2, highlighted in yellow (lines 68-71, 202-204).

Reviewer 2 Report

Comments and Suggestions for Authors

In the work, „The Role of Pyk2 Kinase in Glioblastoma Progression and Therapeutic Targeting“, the authors “Lilia Kucheryavykh and Yuriy Kucheryavykh“, present a nice and informative article; the work is impressive and concise, however, some aspects might still be improved.

In line 91, you write “Pharmacological inhibition of Pyk2 reduced basal migration in U87 and HS683 glioma cells, but had little to no effect on A172, GL261, or C6 cells, suggesting cell line-specific dependencies on Pyk2 signaling”. Please tell what C6 cells are.

And please inform more precisely, which effects of microglia on migration are mediated by Pyk2 and which are independent direct effects of cytokines on glioma cells in the described model.

Line 189: Did Pharmacological inhibition of Pyk2/FAK signaling using PF-562271 also influence proliferation?

Line 252: The authors report that PF-562271 enhances temozolomide – what is the mechanism?

It would be interesting to know about the role of cyclin dependent kinase inhibitors in this context, is there anything published?

Line 349: regarding the peculiar sex-specific and ethnicity-dependent effects that are described – are there hypotheses on the possible cause? Are there really defined sex or ethnicity dependent genes that may shape Pyk2-driven tumor biology?

Is something known about the role of Pyk2 normal brain? Or in other tissues?

Author Response

Comments 1: In line 91, you write “Pharmacological inhibition of Pyk2 reduced basal migration in U87 and HS683 glioma cells, but had little to no effect on A172, GL261, or C6 cells, suggesting cell line-specific dependencies on Pyk2 signaling”. Please tell what C6 cells are.

Response 1: C6 cells are a rat glioma cell line, and we have included this information for clarity, highlighted in yellow (lines 93-94).

Comments 2: And please inform more precisely, which effects of microglia on migration are mediated by Pyk2 and which are independent direct effects of cytokines on glioma cells in the described model.

Response 2: PDGFB, EGF, and SDF-1α all promoted glioma cell migration in a Pyk2-dependent manner (lines 156-157).  Beyond Pyk2, other signaling molecules are implicated in glioma pathogenesis. For instance, (IGFBP-3), Cullin1 (Cul1), (GOLPH3), and (AQP1) have all been associated with glioma progression and invasiveness (lines 111-120).

Comments 3: Line 189: Did Pharmacological inhibition of Pyk2/FAK signaling using PF-562271 also influence proliferation?

Response 3: Yes, we add clarification in the text (lines 200-202), highlighted in yellow.

Comments 4: Line 252: The authors report that enhances temozolomide – what is the mechanism?

Response 4: Although the precise mechanism remains unclear, this article proposes a potential role for Pyk2 in regulating the S/G2 cell cycle transition. Temozolomide (TMZ) induces G2/M arrest as part of a treatment resistance mechanism. We suggest that PF-562271 enhances this arrest by preventing cell cycle re-entry, thereby amplifying the cytotoxic effect of TMZ. (lines 289-296).

Comments 5: It would be interesting to know about the role of cyclin dependent kinase inhibitors in this context, is there anything published?

Response 5: We appreciate the reviewer’s comment regarding the potential role of CDK inhibitors in this context. While this is indeed an important area of investigation, it falls outside the scope of the current study, which focuses specifically on the role of Pyk2 and its modulation by PF-562271 in GBM. We agree that exploring the involvement of CDK inhibitors would be valuable for future research.

Comments 6: Line 349: regarding the peculiar sex-specific and ethnicity-dependent effects that are described – are there hypotheses on the possible cause? Are there really defined sex or ethnicity dependent genes that may shape Pyk2-driven tumor biology?

Response 6: While the exact mechanisms remain to be fully elucidated, we propose the influence of sex hormones, particularly estrogen and androgens, which are known to modulate signaling pathways relevant to glioma progression. These hormonal differences may influence the expression or activity of Pyk2 and its downstream effectors, potentially explaining the sex-specific correlation patterns, such as the exclusive association between Pyk2 and CXCR1 observed in Puerto Rican men. We added this to the manuscript for better clarity (lines 367-370).

Comments 7: Is something known about the role of Pyk2 normal brain? Or in other tissues?

Response 7: We added this for clarity in lines 68-71, and 203-205, highlighted in yellow.

Reviewer 3 Report

Comments and Suggestions for Authors

The manuscript "The Role of Pyk2 Kinase in Glioblastoma Progression and Therapeutic Targeting" sent to the Journal "Cancers". This is an original and interesting study analyzing the effect of Pyk2 on glioblastoma migration and proliferation. The authors structured the paper well, covering important points to clarify the roles of Pyk2 in various aspects of glioblastoma development, as well as resistance to temozolamide treatment among other aspects. However, I consider that it is not ready for publication, it requires major modifications, as I mention below.

  1. The authors should improve figure 2, to make the image more didactic and attractive.
  2. “3.3. Pyk2 and Glioma Proliferation”. In this section they should add references.

  1. These findings are consistent with broader cancer research. For instance, in osteosarcoma xenograft models, PF-562271 treatment reduced tumor weight and volume [55]”. The description in this paragraph does not align with the reference. The authors should carefully review each of the articles they refer to.

  1. The authors describe Pyk2 in people from Puerto Rico, but I consider it important that they talk about other ethnicities, and if there is no information, they should emphasize that point.

  1. I suggest that the authors create a figure that summarises all the Pyk2 effects described in the different subtopics of the paper.

Author Response

Comments 1: The authors should improve figure 2, to make the image more didactic and attractive.

Response 1: We have revised Figure 2.

Comments 2: “3.3. Pyk2 and Glioma Proliferation”. In this section they should add references.

Response 2: We have revised this section for clarity and added additional references, which are highlighted in yellow.

Comments 3: “These findings are consistent with broader cancer research. For instance, in osteosarcoma xenograft models, PF-562271 treatment reduced tumor weight and volume [55]”. The description in this paragraph does not align with the reference. The authors should carefully review each of the articles they refer to.

Response 3: Thank you. The correct reference (57) has been inserted, highlighted in yellow.

Comments 4: The authors describe Pyk2 in people from Puerto Rico, but I consider it important that they talk about other ethnicities, and if there is no information, they should emphasize that point.

Response 4: We emphasized this point in lines 383-386.

Comments 5: I suggest that the authors create a figure that summarises all the Pyk2 effects described in the different subtopics of the paper.

Response 5: Thank you for your suggestion. We have revised Figure 2, and together with Figure 1, we believe it now better illustrates the role of Pyk2 kinase.

Round 2

Reviewer 3 Report

Comments and Suggestions for Authors

No comments